# Decentralized Collaborative Learning with Adaptive Reference Data for On-Device POI Recommendation

## ABSTRACT

In Location-based Social Networks (LBSNs), Point-of-Interest (POI) recommendation helps users discover interesting places. There is a trend to move from the conventional cloud-based model to on-device recommendations for privacy protection and reduced server reliance. Due to the scarcity of local user-item interactions on individual devices, solely relying on local instances is not adequate. Collaborative Learning (CL) emerges to promote model sharing among users. Central to this CL paradigm is reference data, which is an intermediary that allows users to exchange their soft decisions without directly sharing their private data or parameters, ensuring privacy and benefiting from collaboration. While recent efforts have developed CL-based POI frameworks for robust and privacy-centric recommendations, they typically use a single and unified reference for all users. Reference data that proves valuable for one user might be harmful to another, given the wide range of user preferences. Some users may not offer meaningful soft decisions on items outside their interest scope. Consequently, using the same reference data for all collaborations can impede knowledge exchange and lead to sub-optimal performance. To address this gap, we introduce the Decentralized Collaborative Learning with Adaptive Reference Data (DARD) framework, which crafts adaptive reference data for effective user collaboration. It first generates a desensitized public reference data pool with transformation and probability data generation methods. For each user, the selection of adaptive reference data is executed in parallel by training loss tracking and influence function. Local models are trained with individual private data and collaboratively with the geographical and semantic neighbors. During the collaboration between two users, they exchange soft decisions based on a combined set of their adaptive reference data. Our evaluations across two real-world datasets highlight DARD's superiority in recommendation performance and addressing the scarcity of available reference data.

## 1 INTRODUCTION

In recent years, location-based social networks (LBSNs) like Yelp, and Foursquare [6] have become more significant in e-commerce [21, 27]. Within LBSNs, users share their physical locations and experiences with check-in data. Utilizing this check-in data with Point-of-Interest (POI) recommendations is essential to aid users in discovering new POIs, and enhance location-based services, such as mobile advertisements [20]. It is common practice to rely on a powerful cloud server, which not only hosts all user data but also manages the training and inference of the recommendation model [22]. However, this approach raises user privacy concerns since personal check-in histories are frequently shared with service providers. Such centralized recommendations are at risk of violating new privacy regulations (e.g., the General Data Protection Regulation (GDPR)[1]). Additionally, the system's reliance on server capabilities and stable internet connections [33] compromises the service reliability of cloud-based POI recommendations.

This has driven on-device POI recommendation systems [9, 25, 33], which deploy models to edge devices (e.g., smartphones and smart cars), enabling recommendations to be generated locally with minimal reliance on centralized resources. Considering the limited memory capacities of devices in contrast to abundant cloud servers, there's an essential push towards memory compression techniques [1, 5, 10]. One popular strategy is on-device deployment [5, 10], including embedding quantization [38] and the "student-teacher" framework [33], which deploys the same compact model derived from a sophisticated cloud model for all user devices. However, such techniques neglect diverse user interests, and the varied capabilities of devices. As an alternative, on-device learning [2, 25] actively involves each user in the model training process, which utilizes the computational power of devices and crafts personalized models tailored to individual user preferences and device constraints. Given the scarcity of local user-item interactions, rather than solely relying on local instances, Collaborative Learning (CL) [9, 39] has been introduced for on-device POI recommendation, promoting model sharing between users, and avoiding the need for complex cloud-based model designs.

Collaborative Learning (CL) for recommendation has been segmented into two approaches: centralized CL-based recommendation represented by federated learning-based methods [9, 23], and decentralized CL-based recommendation [25, 39]. In the federated recommendation, users train their models locally for data privacy, and a central server is continuously engaged to aggregate these models to counter data sparsity, then distribute an aggregated version back to users. On the other hand, for decentralized CL-based recommendations, the central server's primary role is limited to an initial phase, providing users with pretrained model parameters, specifically POI embeddings, and grouping similar users. Subsequently, these user-specific models are refined through a combination of local training and communication with nearby users within the same group [39]. Intra-group collaboration cloud be facilitated by directly exchanging gradients or raw model parameters between users, assuming identical local model architectures [9, 23]. However, this approach compromises communication efficiency and user privacy by revealing users' sensitive data. Furthermore, the assumption of model homogeneity significantly limits the applicability of such decentralized POI recommendation methods. In real-world scenarios, on-device models often require unique structures tailored to individual device capacities (e.g., memory budget and computation resources) [33]. Therefore, a dataset called "reference data" is introduced as an intermediary for user collaboration across heterogeneous models. Reference data serves as a set of data points, aiding in communication among users. Essentially, it functions as a medium that allows users to exchange their soft decisions without directly sharing their private data. This indirect method of knowledge exchange through reference data preserves user privacy while still benefiting from collaborative model refinement. Users interpret and respond to the reference data based on their individual

[1]https://gdpr-info.eu/

preferences and histories, enabling a collaborative and personalized learning experience.

However, existing decentralized CL-based recommendation systems [2, 26, 39] commonly utilize a uniform reference dataset for all users, often neglecting spatial dynamics and user preference diversity. Prompted by the underlying question — do different users necessitate tailored reference datasets? We undertake a preliminary investigation using the Foursquare dataset [6]. The experimental setting is introduced in Appendix A. Following [26], we derive the desensitized public candidate pool from the private check-in sequence and compose different sets of reference data. These reference data are chosen at random, based on POI popularity, or adaptively selected for individual users. For the collaboration between two users, two users will share and align their soft decisions on the joint set of their selected reference data. Figure 1 depicts the performance for a typical POI recommendation system STAN [27] in the CL paradigm, measured via the hit ratio, when these diverse reference dataset selections are employed. Two clear conclusions can be made: (i) Adapting the same original candidate pool for all users is sub-optimal. Users' local models are unlikely to provide accurate predictions on distant items or items out of their interest, which runs the risk of introducing noise, thereby hampering the recommendation quality. (ii) The selection of reference data markedly impacts recommendation results. Both the adaptive reference data and data stemming from popular POI items demonstrate better performance than random selection. It becomes apparent that data closely aligned with user preferences enhances the model's effectiveness. Unlike traditional CL tasks, such as decentralized computer vision model training [30, 32] that employs a shared public reference dataset, POI recommendation presents unique challenges. In conventional tasks, each local model shares a uniform data distribution (e.g., flowers) and pursues a common objective (e.g., classifying flower types). However, in POI recommendations, users exhibit varied data distributions due to their distinct interests [31]. The aim of each model is to deliver personalized recommendations for its specific user. Consequently, reference data that proves valuable for one user might be harmful to another, given the wide range of user preferences. Some users may not offer meaningful soft decisions on items outside their interest scope, emphasizing the vital need for a CL recommendation paradigm with an adaptive reference dataset.

To this end, we introduce a novel framework: Decentralized Collaborative Learning with Adaptive Reference Data (DARD). It involves a server that identifies user clusters based on similarities, subsequently enabling collaborative learning among neighbors. The core of this adaptive approach is an innovative Knowledge Distillation (KD) mechanism, designed to facilitate seamless knowledge exchange via soft decisions on a combined set of their adaptive reference data. Notably, the adoption of soft decisions on adaptive reference data avoids the traditional requirement that models within the CL paradigm remain identical, paving the way for greater real-world applicability, where varying devices (e.g., mobile phones or Internet-of-Things [42]) possess diverse memory capacities or model architectures. Concerning the availability of public reference data and user privacy, many current methodologies directly allocate a portion of private historical check-in data for public use. To

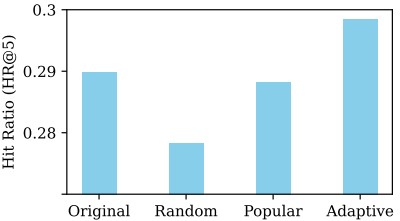

**Figure 1: Impact of reference data selection on recommendation performance. "Original" refers to no selection, whole reference data candidate pool for all users.**

address this, we utilize two check-in data generation methods: transformation and probability generation, ensuring robust reference data candidate pools while safeguarding user privacy.

A straightforward approach to identifying this adaptive reference data from the public reference data candidate pool would entail retraining the CL paradigm using all possible combinations from a public reference dataset. Such an approach is clearly infeasible due to its computational demands. To resolve this, we first track individual training loss to delete the noisy instances with large losses from the candidate pool during the training for every user and then utilize the influence function to estimate the influence of each instance after convergence. adaptive reference data for individual users are filtered out on the devices in parallel. Subsequently, the model is retrained within the CL paradigm using the adaptive reference data. In contrast to exhaustive retraining, DARD offers a one-time selection of optimal adaptive reference data.

The contributions of this paper are summarized as follows:

- The introduction of the DARD framework, designating heterogeneous models to engage in knowledge exchange with soft decisions on adaptive reference data. It accommodates different on-device recommenders, tailors for distinctive user preferences, and lessens the dependency on servers.
- The introduction of the training loss tracking and influence function on the devices to select adaptive reference data for individual users.
- A comprehensive evaluation of DARD using two real-world datasets, emphasizing its effectiveness in recommendation performance, and addressing the scarcity of available reference data.

## 2 RELATED WORK

This section will review the main related works including on-device recommender systems and decentralized collaborative learning (CL) recommender systems.

### 2.1 On-device Recommender Systems

Unlike cloud-based recommendations [16, 29, 41], where models operate on the cloud, on-device recommender systems transfer the model processing from the cloud directly to the device. This paradigm encompasses two main approaches: (1) On-device deployment [33, 38]: model is fully trained on the cloud server and subsequently deployed solely on the device. Techniques like embedding quantization [38] and the "student-teacher" framework [33] deploy a compact version of a sophisticated cloud model across all user devices. (2) On-device learning [2, 9]: model is trained directly on the device. Given the limited local user-item interactions, Collaborative

Learning (CL) [30, 32] is often utilized. CL-based methods can be split into centralized and decentralized techniques. In the centralized CL, federated recommendation [9, 23] has gained prominence: users train their local models while a central server aggregates these models to address data sparsity and then redistributes the combined model to users. In contrast, decentralized CL [2, 26, 39] limits the central server's role to the initial phase, supplying users with pretrained model parameters. These user-specific models are then enhanced by a mix of local training and interaction with users within the same cluster.

## 2.2 Decentralized Collaborative Learning (CL) Recommender Systems

Decentralized CL-based recommendations [2, 4, 25, 26, 39] depend on the server only during the initial stage, primarily to obtain pretrained models and to determine the neighbor set. Following this, user-specific models are first trained locally using private data, and then collaboratively with neighbors. For knowledge exchange, raw parameters or gradients are shared directly [4, 25]. With a heightened emphasis on privacy and ensuring collaboration between heterogeneous models, reference data is introduced as an intermediary. Instead of direct parameter exchange, users only share their soft decisions made on this reference data. D-Dist [2] allows local models to communicate with randomly chosen heterogeneous neighbors, and SQMD [39] determines neighbors based on shared responses to the reference dataset. MAC [26] prioritizes communication via a public reference dataset and prunes non-essential neighbors during training. Since the reference data facilitates user collaboration and knowledge exchange, the selection of reference data is vital. However, existing methods neglect it, and utilize one single reference data for all users.

## 3 PRELIMINARY

In this section, we first introduce some important definitions frequently used in POI recommendation and problem formulation.

**Definition 1: Check-in Sequence** A check-in sequence $\mathcal{X}(u_j) = \{p_1, p_2, \cdots, p_{n_j}\}$ contains $n_j$ chronologically visited Point-of-Interet (POI) by the user $u_j \in \mathcal{U}$, where POI data $p \in \mathcal{P}$.

**Definition 2: Category Sequence** For user $u_j$, a category sequence $\mathcal{X}^c(u_j) = \{c_{p_1}, c_{p_2}, \cdots, c_{p_{n_j}}\}$ includes the corresponding category $c_{p_j}$ for POI $p_j$, and $C$ represents the set for all categories.

**Definition 3: Geographical Segment (Region).** A region $r$ represents a geospatial division of POIs. Following [25, 26], we derive a collection of regions $\mathcal{R}$ by performing $k$-means clustering [28] on the geographical coordinates of all POIs in this study.

**Definition 4: Public Reference Candidate Pool**. A public reference candidate pool $\mathcal{D} = \mathcal{D}^g \cup \mathcal{D}^s$, comprises both geographical reference data $\mathcal{D}^g$ and semantic reference data $\mathcal{D}^s$. These cater to collaborative processes among geographical and semantic neighbors, respectively. The methodologies for formulating $\mathcal{D}^g$ and $\mathcal{D}^s$ are elucidated in Section 4.1. Concerning a particular user $u_j$, an adaptive reference data, designated as $\hat{\mathcal{D}}(u_j) \subseteq \mathcal{D}$, is selected from $\mathcal{D}$ and discussed in Section 4.3.

**Problem Formulation: Decentralized CL POI Recommendation with Adaptive Reference Data**. In the DRAD framework, we assign specific roles to the devices/users and the central server as detailed below:

- **Device/User Role**: An individual user $u_j$ possesses her distinct check-in sequences $\mathcal{X}(u_j)$, category sequences $\mathcal{X}^c(u_j)$, and a tailor-made model $\phi_j(\cdot)$ that is collectively trained using local data and further refined via user collaborations. For efficient storage, model $\phi_j(\cdot)$ only retains embeddings corresponding to POIs within regions the user has visited or is currently in $r \in \mathcal{R}(u_j)$. $\mathcal{D}$ is stored on the device and then replaced by adaptive reference data $\hat{\mathcal{D}}(u_j)$.
- **Server Role**: The server's main task is to determine neighbor sets for all users with the collected low-sensitivity data and to generate the reference data candidate pool. Once the server sends this data to users, it remains uninvolved during the subsequent local model training phase.

The adaptive reference data $\hat{\mathcal{D}}(u_j)$ is selected on each user $u_j$ device from the public reference data pool $\mathcal{D}$. The on-device model $\phi_j(\cdot)$ is then leveraged to predict a prioritized list of potential POIs for the user's next movement.

## 4 DARD

This section introduces DARD framework, where an overview is depicted in Figure 2. The main components include: (1) Reference data candidate pool generation. (2) Model collaborative learning paradigm for both Step 2 model training and Step 4 model retraining. (3) Select adaptive reference data for an individual user based on its own training loss during training, and influence function after convergence. (4) Model retraining under the CL paradigm with adaptive reference data.

### 4.1 Reference Data Candidate Pool Generation

Inspired by [26], for each user $u_i$, there are two methods to generate non-sensitive check-in and categories sequences locally and upload them to the cloud server to aggregate the reference data candidate pool $\mathcal{D} = \mathcal{D}^g \cup \mathcal{D}^s$, where $\mathcal{D}^g(u_i) = \{\mathcal{X}_v\}_{v=1}^V$ is the geographic reference data and $\mathcal{D}^s(u_i) = \{\mathcal{X}_z^c\}_{z=1}^Z$ is the semantic reference data. The neighbor identification is universal to all CL-based recommendations. We introduce it in Appendix C.1, since our contribution lies in the reference data.

**Transformation Generation**: Instead of directly revealing users' check-in sequences, for Sequence $\{p_1, \cdots, p_i, \cdots, p_n\}$ and $\{p_2, \cdots, p_i, \cdots, p_m\}$ we generate the desensitized check-in sequence by exchanging sequences after the same POI (i.e., $p_i$) to form $\{p_1, \cdots, p_i, \cdots, p_m\}$ and $\{p_2, \cdots, p_i, \cdots, p_n\}$. For all these new sequences, their region(s) $r$ is defined based on the most frequently visited locations, excluding POIs outside region $r$ to ensure region-specific sequences. If users do not want to reveal any specific POIs, the probability generation method generates check-in sequences only with category-level sequences.

**Probability Generation**: For semantic neighbors who might be geographically distant, probability generation relies exclusively on statistics derived from users' non-sensitive category sequences, which are initiated by computing the conditional probabilities for all categories, informed by the aggregated category sequences of users. Specifically, $\mathcal{P}(c_n) = \{P(c_n|c_1), P(c_n|c_2), \cdots, P(c_n|c_{|C|})\}$ encapsulates all conditional probabilities for $c_n$. Each probability $P(c_n|c_m)$ is derived as:

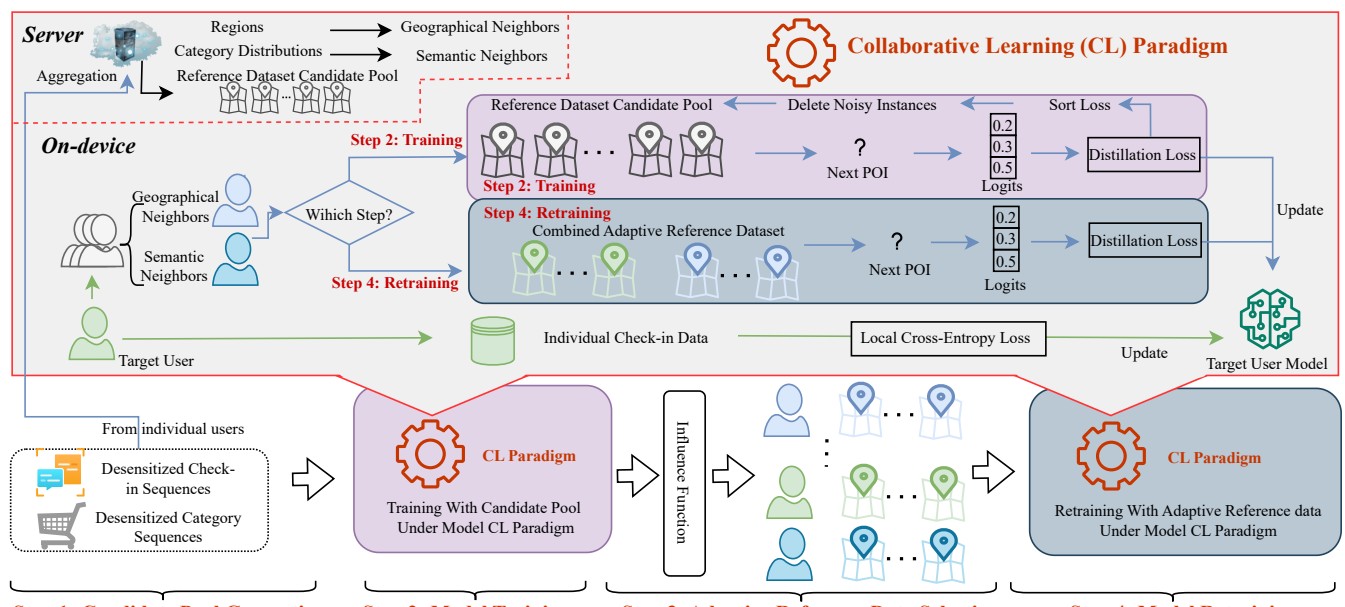

**Figure 2: The overview of DARD. a) Step 1: generate desensitized sequences on-device and upload to server to aggregate as candidate pool. Server only involves in the initial stage to deploy pool and defines neighbors for the user. b) Model CL paradigm: user models are trained with individual data and collaboratively with neighbors. c) Step 2: train under CL paradigm with candidate pool and track loss to delete noisy reference data instances for target user. d) Step 3: utilize influence function to select adaptive reference data. e) Step 4: retrain under CL paradigm with adaptive reference data.**

$$P(c_n|c_m) = \frac{count(c_n|c_m)}{\sum_{m'=1}^{|C|} count(c_n|c_{m'})}, \qquad (1)$$

where $count(c_n|c_m)$ is the total number of $c_m$ following $c_n$. A generated category sequence $\mathcal{X}^c$ emerges by (1) randomly choosing a category as the initiator; and (2) iteratively determining subsequent categories based on the preceding category's conditional probabilities. Through iterative generation of numerous $\mathcal{X}^c$ sequences, a comprehensive semantic reference data candidate pool $\mathcal{D}^s$ encompassing all categories is created. Subsequently, for each specific region, we generate sequences of POIs aligned with the category sequence $\mathcal{X}^c \in \mathcal{D}^s$, further imposing a 5km distance constraint between consecutive POIs.

## 4.2 Collaborative Learning Paradigm

DARD utilizes the general model collaborative learning (CL) paradigm both for training and retraining, as shown in Figure 2, where models are trained with private data on local cross-entropy loss and collaboratively with others on distillation loss. The difference is that, for reference data $\mathcal{D}(u_i)$ in the collaboration procedure, Step 2 training uses the candidate pool $\mathcal{D}(u_i) = \mathcal{D}$, but Step 4 retraining uses the adaptive reference data $\mathcal{D}(u_i) = \hat{\mathcal{D}}(u_i)$.

*4.2.1 Local Cross-entropy Loss Function.* $\mathcal{X}(u_i)$ is the private check-in data, and the local loss function is defined as [26]:

$$L_{loc}(u_i) = l\left(\phi_i\left(\mathcal{X}(u_i)\right), \mathcal{Y}(u_i)\right), \qquad (2)$$

where $l$ stands for the cross-entropy loss function [8] in our case, and $\phi_i(\cdot)$ is the model for user $u_i$. The model is optimized through the local loss function on the private check-in data.

*4.2.2 Distillation Loss Function.* For user $u_i$, $\mathcal{D}(u_i) = \mathcal{D}^g(u_i) \cup \mathcal{D}^s(u_i)$ consists geographical and semantic reference data. We employ the distillation loss to learn insights from soft decisions shared by geographical neighbors $\mathcal{G}(u_i)$ and semantic neighbors $\mathcal{S}(u_i)$.

**Collaboration with Geographical Neighbors**. For user $u_i$, the CL with geographical neighbors $u_j$ is achieved by reducing their difference in soft decisions over $\mathcal{D}^g(u_i) \cup \mathcal{D}^g(u_j)$. The distillation loss is measured as:

$$L_{geo} = \frac{1}{|\mathcal{G}(u_i)|} \sum_{u_j \in \mathcal{G}(u_i)} \left( \sum_{\mathcal{X} \in \mathcal{D}^g(u_i) \cup \mathcal{D}^g(u_j)} \left\| \phi_i\left(\mathcal{X}\right) - \phi_j(\mathcal{X}) \right\|_2^2 \right), \qquad (3)$$

where $\phi_i(\cdot)$ and $\phi_j(\cdot)$ are the local recommendation models for $u_i$ and her neighboring users. $||\cdot||_2^2$ is the normalization term.

**Collaboration with Semantic Neighbors**. Similar to the geographical neighbor collaboration, we align the soft decisions of $u_i$ and her semantic neighbor $u_j \in \mathcal{S}(u_i)$ on the join set of their semantic reference data $\mathcal{D}^s(u_i) \cup \mathcal{D}^s(u_j)$ with distillation loss:

$$L_{sem} = \frac{1}{|\mathcal{S}(u_i)|} \sum_{u_j \in \mathcal{S}(u_i)} \left( \sum_{\mathcal{X}^c \in \mathcal{D}^s(u_i) \cup \mathcal{D}^s(u_j)} \left\| \phi_i\left(\mathcal{X}^c\right) - \phi_j(\mathcal{X}^c) \right\|_2^2 \right), \qquad (4)$$

Therefore, the final loss function for the collaborative learning paradigm is the combination of local loss and collaboration loss, where $\gamma$ and $\mu$ control the preference for individual components:

$$L_{total} = L_{loc} + \gamma L_{geo} + \mu L_{sem}. \qquad (5)$$

## 4.3 Adaptive Reference Data Selection

In this section, we propose to track the loss function during the Step 2 training, and utilize the influence function after training in Step 3 to identify the harmful data instances and select adaptive reference data for individual users, which is conducted on the device side in parallel without the reliance on the cloud server.

*4.3.1 Loss Tracking During Training.* For each user, we track the training loss during her collaboration with semantic (or geographical ) neighbors to identify harmful instances in the semantic (or geographical) reference data, respectively. Both geographical and semantic data instances require refinement. While data selection for each user occurs concurrently, we focus on the selection process for a single user. For notation simplicity, we omit the neighbor notation and represent individual reference data with $\mathcal{D}$.

For a given $m^{th}$ mini-batch of training instances, symbolized as $\bar{\mathcal{D}}^m \in \mathcal{D}$. DARD processes every sample in $\bar{\mathcal{D}}^m$, and sequences them based on their training losses. Large loss instances are regarded as noisy, while their counterparts with small losses are tagged as "clean", denoted by $\bar{\mathcal{D}}_+^m$:

$$\bar{\mathcal{D}}_+^m = \arg \min_{\bar{\mathcal{D}}^m : |\bar{\mathcal{D}}^m| \geq \rho |\bar{\mathcal{D}}^m|} L(\bar{\mathcal{D}}^m, \theta), \tag{6}$$

where $\theta$ is the set of the model of the target user and her neighbors, $\rho$ is the selection ratio, and $L(\cdot)$ is the distillation loss $L_{geo}$ in Eq. 3 or $L_{sem}$ in Eq. 4, depending on neighbor types. For $m^{th}$ mini-batch, the noisy instances are recorded in the set $\bar{\mathcal{D}}_-^m = \bar{\mathcal{D}} \setminus \bar{\mathcal{D}}_+^m$. The adaptive reference data is selected as $\mathcal{D}' = \mathcal{D} \setminus \{\bar{\mathcal{D}}_-^m\}_{m=1}^M$.

*4.3.2 Influence Function After Training.* After Step 2 model training is finished, given adaptive reference data $\mathcal{D}'$, we further examine the contribution of each data instance over the model's performance. A naive method is leave-one-out retraining, which deletes one instance and retrains the model to record the performance difference between two models. Such an approach is clearly infeasible due to its computational demands. Instead, influence functions, stemming from Robust Statistics [12] have been provided as an efficient way to estimate how a small perturbation of a training sample would change the model's predictions [14, 40]. Let $l(\mathcal{X}_j, \theta)$ be loss on instance $\mathcal{X}_j$. For notation simplification, we omit the user index and use $l_j(\theta)$ to present $l(\mathcal{X}_j, \theta)$. Considering the standard empirical risk minimization (ERM) as the optimization objective, the empirical risk is defined as $L(D; \theta) = \frac{1}{|\mathcal{D}'|} \sum_{i=1}^{|\mathcal{D}'|} l_i(\theta)$.

let $\hat{\theta} = \arg \min_\theta \frac{1}{|\mathcal{D}'|} \sum_{j=1}^{|\mathcal{D}'|} l_j(\theta)$ be the optimal model parameters. when upweighing a training instance $\mathcal{X}_j$ by an infinitesimal step $\epsilon_j$ on its loss term, we could acquire the new optimal parameters: $\hat{\theta}_{\epsilon_j} = \arg \min_\theta \frac{1}{|\mathcal{D}'|} \sum_{j=1}^{|\mathcal{D}'|} l_j(\theta) + \epsilon_j l_j(\theta)$. Based on influence functions [13, 15], we have the following expression to estimate the changes of the model parameters when upweighting $\mathcal{X}_j$ by $\epsilon_j$:

$$\psi_\theta(\mathcal{X}_j) = \frac{\partial \hat{\theta}_{\epsilon_j}}{\partial \epsilon_j} |_{\epsilon_j=0} = -H_{\hat{\theta}}^{-1} \nabla_\theta l_j(\hat{\theta})$$

$$H_{\hat{\theta}} = \frac{1}{|\mathcal{D}'|} \sum_{j=1}^{|\mathcal{D}'|} \nabla_\theta^2 l_j(\hat{\theta}) \tag{7}$$

where $H_{\hat{\theta}}$ is the Hessian matrix and $\nabla_\theta^2 l_j(\hat{\theta})$ is the second derivative of the loss at the training instance $\mathcal{X}_j$ with respect to $\theta$. After applying the chain rule, we are able to estimate the changes in model prediction at the validation instance $\mathcal{X}_k^v$:

$$\Psi_\theta(\mathcal{X}_j, \mathcal{X}_k^v) = \frac{\partial l_k(\hat{\theta}_{\epsilon_j})}{\partial \epsilon_j} |_{\epsilon_j=0} = -\nabla_\theta l_j(\hat{\theta}) H_{\hat{\theta}}^{-1} \nabla_\theta l_j(\hat{\theta}) \tag{8}$$

Assume we have a validation set $Q = \{\mathcal{X}_1^v, \mathcal{X}_2^v, \cdots, \mathcal{X}_{n'}^v\}$ to evaluate the performance of the model after collaborative learning between neighbors by exchanging soft labels on the reference data. we compute $\mathcal{D}_-' \subseteq \mathcal{D}'$ which contains harmful training samples. A training sample is harmful to the model performance if removing it from the training set would reduce the test risk over $Q$. Based on influence functions, we can measure one sample's influence on test risk without prohibitive leave-one-out retraining. According to Eq. (7) (8), if we add a small perturbation $\epsilon_j$ on the loss term of $\mathcal{X}_j$ to change its weight, the change of test loss at a validation instance $\mathcal{X}_k$ can be estimated as follows:

$$l(\mathcal{X}_k^v, \hat{\theta}_{\epsilon_j}) - l(\mathcal{X}_k^v, \hat{\theta}) \approx \epsilon_j \times \Psi_\theta(\mathcal{X}_j, \mathcal{X}_k^v) \tag{9}$$

where $\Psi_\theta(\cdot, \cdot)$ is computed by Eq. (8). We then estimate the influence of perturbing $\mathcal{X}_j$ on the whole test risk as follows:

$$l(Q, \hat{\theta}_{\epsilon_j}) - l(Q, \hat{\theta}) \approx \epsilon_j \times \sum_{k=1}^{n'} \Psi_\theta(\mathcal{X}_j, \mathcal{X}_k^v) \tag{10}$$

Henceforth, we denote by $\Psi_\theta(\mathcal{X}_j) = \sum_{k=1}^{n'} \Psi_\theta(\mathcal{X}_j, \mathcal{X}_k^v)$ the influence of perturbing the loss term of $\mathcal{X}_j$ on the test risk over $Q$. It is worth mentioning that given $\epsilon_j \in [-\frac{1}{|\mathcal{D}'|}, 0)$, Eq. (10) computes the influence of discarding or downweighting the instance $\mathcal{X}_j$. We denote $\mathcal{D}_-' = \{\mathcal{X}_j \in \mathcal{D}' | \Psi_\theta(\mathcal{X}_j) > 0\}$ as harmful samples. In contrast to the leave-one-out strategy which requires $|\mathcal{D}'|$ (i.e., size of reference data) times retraining, DARD offers a one-time selection of optimal adaptive reference data. Similar to [15, 36], we assume that each training sample influences the test risk independently. We derive the Lemma 1 and the proof is provided in Appendix B.

LEMMA 1. *Discarding or downweighting the training samples in $\mathcal{D}_-' = \{\mathcal{X}_j \in \mathcal{D}' | \Psi_\theta(\mathcal{X}_j) > 0\}$ from $\mathcal{D}'$ could lead to a model with lower test risk over $Q$:*

$$L(Q, \hat{\theta}_\epsilon) - L(Q, \hat{\theta}) \approx -\frac{1}{m} \sum_{\mathcal{X} \in \mathcal{D}_-'} \Psi_\theta(\mathcal{X}_j) \tag{11}$$

*where $\hat{\theta}_\epsilon$ denotes optimal model parameters obtained by updating parameters with discarding or downweighting samples in $\mathcal{D}_-$.*

Lemma 1 elucidates why identifying harmful instances and selecting beneficial adaptive reference data, denoted as $\hat{\mathcal{D}}'(u_i) = \mathcal{D}'(u_i) \setminus \mathcal{D}_-'(u_i)$ for each user $u_i$, is feasible and essential for DARD. Instead of downweighting the harmful instances, we choose to directly exclude them to save the device budget. To account for potential estimation errors in $\Psi_\theta(\mathcal{X}_j)$, which could misidentify harmful training samples, we define $\mathcal{D}_-'(u_i) = \{\mathcal{X}_j \in \mathcal{D} | \Psi_\theta(\mathcal{X}_j) > \alpha\}$, where hyper-parameter $\alpha$ is further studied in Section 5.4.

**Algorithm 1:** Optimizing DARD. Processes are implemented on device side.

---

**1** **foreach** $u_i \in \mathcal{U}$ **in parallel do**

**2**   Receive $\mathcal{G}(u_i), S(u_i), \mathcal{D}(u_i)$ ;

  // Train under CL paradigm and loss tracking

**3**   **for** $n = 1$ *to* $N$ **do**

**4**    Receive soft decisions from $\mathcal{G}(u_i), \mathcal{S}(u_i)$;

**5**    **for** $m = 1$ *to* $M$ **do**

**6**     Fetch $m$-th mini-batch $\bar{\mathcal{D}}^m(u_i)$ from $\mathcal{D}(u_i)$;

**7**     $\bar{\mathcal{D}}^m_+ = \arg\min_{\bar{\mathcal{D}}:|\bar{\mathcal{D}}| \geq \rho|\bar{\mathcal{D}}|} L(\bar{\mathcal{D}}, \theta)$;

**8**     $\bar{\mathcal{D}}^m_- = \bar{\mathcal{D}}^m \setminus \bar{\mathcal{D}}^m_+$;

**9**     $L_{total} = L_{loc} + \gamma L_{geo}(\bar{\mathcal{D}}_+) + \mu L_{sem}(\bar{\mathcal{D}}_+)$;

**10**     Update the model: $\phi_i \leftarrow \phi_i - \eta \nabla L_{total}$;

**11**   $\mathcal{D}'(u_i) = \mathcal{D}(u_i) \setminus \{\mathcal{D}^m_-\}^M_{m=1}$;

  // Select data with influence function

**12**   **for** $j = 1$ *to* $|\mathcal{D}'(u_i)|$ **do**

**13**    Calculate the influence of the reference data instance $\mathcal{X}_j \in \mathcal{D}'(u_i)$ using Eq. (10);

**14**    **if** $\Psi_\phi(\mathcal{X}_j) \geq \alpha$ **then**

**15**     $\mathcal{D}'(u_i) = \mathcal{D}'(u_i) \setminus \mathcal{X}_j$;

**16**   $\hat{\mathcal{D}}(u_i) = \mathcal{D}'(u_i)$ ;

  // Retrain under CL paradigm with $\hat{\mathcal{D}}(u_i)$

**17**   **for** $n = 1$ *to* $N$ **do**

**18**    Receive soft decisions from $\mathcal{G}(u_i), \mathcal{S}(u_i)$;

**19**    $L_{total} = L_{loc} + \gamma L_{geo}(\hat{\mathcal{D}}(u_i)) + \mu L_{sem}(\hat{\mathcal{D}}(u_i))$;

**20**    Update the model: $\phi_i \leftarrow \phi_i - \eta \nabla L_{total}$;

**21**   Output $\hat{\mathcal{D}}(u_i)$ and $\hat{\phi}_i$;

---

### 4.4 The Optimization Method

We look into the optimization of DARD in this section. The cloud server only participates in the initial stage, where it aggregates the desensitized sequences locally generated by users to form the reference data candidate pool, identify neighbors, and deploy the candidate pool to the individual users. All the processes are executed on the device side as shown in Algorithm 1. First, users are trained with local data and with neighbors under CL paradigm (lines 3-11), where noisy data is identified by loss tracking. After training, the refined reference data $\mathcal{D}'$ is further examined by influence function to isolate the harmful instances by estimating the performance difference (lines 12-16). At last, the model is retrained under CL paradigm with adaptive reference data $\hat{\mathcal{D}}(u_i)$ to output the personalized recommendation model $\hat{\phi}_i(\cdot)$.

## 5 EXPERIMENTS

To validate the effectiveness of the proposed method, we perform comprehensive experiments to respond to the following research questions (RQs):

- **RQ1:** How does our proposed method compare against existing centralized and decentralized recommendation methods?
- **RQ2:** How effective is our method, especially in situations where the reference data is limited?

- **RQ3:** How do various hyper-parameters impact the performance of the proposed method?
- **RQ4:** How do individual components within our method influence its overall performance?
- **RQ5:** Can the proposed method be incorporated into different CL-based recommendation approaches?

**Table 1: The statistics of datasets.**

|  | #users | #POIs | #check-ins | #check-ins per user | # categories |
|---|---|---|---|---|---|
| Weeplace | 4,560 | 44,194 | 923,600 | 202.54 | 625 |
| Foursquare | 7,507 | 80,962 | 1,214,631 | 161.80 | 436 |

### 5.1 Experimental Settings

*5.1.1 Datasets.* We utilize two widely recognized real-world Location Social Network datasets for the assessment of our proposed DARD: Weeplace [24] and Foursquare [6]. Both datasets encompass users' check-in histories in different cities. Following [3, 18], POIs and users with less than 10 interactions are excluded. The key features of these datasets are presented in Table 1.

*5.1.2 Evaluation Protocols.* Following [34, 35], for each check-in sequence, the last check-in POI is for testing, the second last for validation, and the rest for training. Sequences exceeding a length of 200 are truncated to the latest 200 check-ins. In the evaluation phase, instead of evaluating against all items as in [17], each ground truth is ranked against 200 unvisited POIs, located within the same region. This approach recognizes the location-sensitive nature of POI recommendations; users are unlikely to consecutively visit distant POIs [19, 25]. Recommender produces a ranked list of 201 POIs based on scores, with the ground truth ideally ranking highest. Two ranking metrics are used: Hit Ratio at Rank $k$ (HR@$k$) and Normalized Discounted Cumulative Gain at Rank $k$ (NDCG@$k$) [37]. While HR@$k$ focuses on the frequency the ground truth appears in the top-$k$ list, NDCG@$k$ emphasizes its high rank.

*5.1.3 Baselines.* We compared DARD with various recommendation approaches, which include centralized cloud-based methods ( where the model is deployed on the cloud side), centralized on-device methods (where on-device recommendations are performed and a cloud server is heavily involved), and decentralized CL methods (where the server engages primarily during initialization, followed by user collaboration).

**Centralized Cloud Recommendation: MF** [22] is a traditional centralized POI system based on matrix factorization. **LSTM**[11] employs a recurrent neural network to capture the sequential data dependencies. **STAN** [27] discerns spatiotemporal correlations in check-in paths using a bi-attention mechanism.

**Centralized On-device Recommendation:** It refers to methods that deploy a recommendation model on the device, but still heavily rely on a central server. **LLRec** [33] uses a teacher-student strategy to derive a locally deployable compressed model. **PREFER** [9] as a federated POI recommendation paradigm, uses a could server to gather and aggregates locally optimized models, and redistribute the federated model.

**Decentralized CL Recommendation: DCLR** [25] facilitates knowledge sharing among similar neighbors through attentive aggregation and mutual information optimization. **D-Dist** [2] focuses on allowing local models to engage with randomly heterogeneous

Table 2: The top-k recommendation performance of DARD and baselines on two datasets. The best results are marked in bold.

| Category | Method | Weeplace | | | | Foursqaure | | | |
|---|---|---|---|---|---|---|---|---|---|
| | | HR@5 | NDCG@5 | HR@10 | NDCG@10 | HR@5 | NDCG@5 | HR@10 | NDCG@10 |
| Centralized Cloud | MF | 0.1071 | 0.0734 | 0.1323 | 0.0918 | 0.0842 | 0.0624 | 0.0958 | 0.0675 |
| | STAN | 0.3151 | 0.1788 | 0.4570 | 0.2735 | 0.2957 | 0.1710 | 0.4032 | 0.2535 |
| | LSTM | 0.2394 | 0.1382 | 0.3209 | 0.1644 | 0.1954 | 0.1226 | 0.2914 | 0.1663 |
| Centralized On-device | PREFER | 0.2898 | 0.1801 | 0.3644 | 0.2253 | 0.2848 | 0.1619 | 0.3619 | 0.2162 |
| | LLRec | 0.2875 | 0.1723 | 0.3615 | 0.2295 | 0.2767 | 0.1404 | 0.3365 | 0.1826 |
| Decentralized CL | D-Dist | 0.2490 | 0.1270 | 0.3573 | 0.1939 | 0.2227 | 0.1120 | 0.2874 | 0.1667 |
| | DCLR | 0.3281 | 0.1858 | 0.4610 | 0.2708 | 0.3052 | 0.1740 | 0.4291 | 0.2549 |
| | SQMD | 0.2913 | 0.1507 | 0.4306 | 0.2171 | 0.2784 | 0.1433 | 0.4053 | 0.2311 |
| | MAC | 0.3338 | 0.1889 | 0.4786 | 0.2808 | 0.2967 | 0.1735 | 0.4261 | 0.2606 |
| | DARD | **0.3408** | **0.1967** | **0.4863** | **0.2951** | **0.3098** | **0.1798** | **0.4311** | **0.2689** |

neighbors, leveraging their soft decisions based on a shared reference dataset. **SQMD** [39] like D-Dist, operates on a decentralized distillation framework, defining neighbors by their shared reference dataset responses. **MAC** [26] also adopts a decentralized knowledge distillation framework, emphasizing communication based on a public reference dataset while pruning non-essential neighbors during training.

*5.1.4 Hyper-parameters Setting.* Following [25, 26] we utilize STAN as the base model for DARD. For general parameters in decentralized CL recommendations, the number of neighbors is defined as 50, $\gamma$ is 0.5, and $\mu$ is 0.7. We set the dimension to 64, learning rate *eta* to 0.002, dropout to 0.2, batch size M to 16, and training epoch N to 50 for all methods. As the key hyper-parameters for data selection in DARD, $\alpha$ and $\rho$ are set as 0.001 and 0.8. Furthermore, 5% users are randomly selected to generate desensitized check-in data with the same amount of local private data and upload it to the cloud for candidate pool generation. Experimental results are executed five times and averaged.

## 5.2 Top-k Recommendation (RQ1)

To validate the effectiveness of DARD, we compare it with different categories of baselines on the Top-k recommendation task. Table 2 presents recommendation performance results, and our analysis yields several key observations.

Within the centralized POI recommenders category, STAN exhibits superior accuracy compared to LSTM and MF. Despite this, DARD surpasses STAN's performance. A potential reason for STAN's limitations might be its training on check-ins spanning multiple cities. This approach risks integrating knowledge from one region that may be irrelevant or harmful to recommendations in another, thereby reducing STAN's efficacy. Furthermore, DARD consistently achieves better results than centralized on-device methods, where devices rely on the server for the entire time. Instead of collaborating with all users through the server, DARD implements collaboration between similar neighbors to enhance personalization. In contrast, centralized models with a faint cloud model tend to accommodate the majority's preferences, neglecting diverse user interests. Finally, when compared to decentralized CL methods that use a reference dataset, such as D-Dist, SQMD, and MAC, DARD retains its superiority. This can be attributed to DARD's innovative use of

an adaptive reference dataset, which better supports collaborative user learning and more effective knowledge exchange.

## 5.3 Limited Reference Data (RQ2)

To investigate the performance of various decentralized CL methods under limited amounts of reference data, the proportion of data was systematically reduced from 0.8 to 0.1 on the Weeplace dataset. A value of 0.8 indicates that only 80% of the reference data from the cloud candidate pool is transferred to the device for usage. The recommendation performance is evaluated with HR@10 on the Weeplace dataset, as shown in Figure 3.

The diminishing performance of most CL-based methods with reduced reference data underscores the critical role of this data in recommendation quality. In contrast, our proposed DARD method shows resilience, with only minor performance drops from 0.8 to 0.3, highlighting its effective reference data selection capability. This suggests that DARD efficiently identifies key instances vital for knowledge exchange; thus, reducing the on-device reference data doesn't drastically impact knowledge exchange quality. However, DARD's performance does decline when the data portion is cut from 0.3 to 0.1. A possible explanation is that the 0.1 threshold may be too restrictive to encompass all essential instances in the reference data, causing the exclusion of some valuable instances and a subsequent drop in performance. Notably, DARD with reduced reference data (e.g., 0.3) still outperforms other methods using larger data sets. DARD not only yields superior results but also efficiently manages the communication load, given that the communication cost is intrinsically linked to the volume of reference data whose soft decisions are exchanged.

## 5.4 Hyper-parameter Study (RQ3)

To investigate the effects of the key hyper-parameters for adaptive reference data selection, we examine the selection ratio for tracking training loss, $\rho$, ranging between {0.1, 0.2, 0.3, 0.4, 0.5} and influence function parameter, $\alpha$ in the set {0.01, 0.005, 0.001, 0.0005, 0.0001}. We adjusted each $\rho$ and $\alpha$ value individually, maintaining another hyper-parameter constant, and documented the recommendation outcomes by HR@10, depicted in Figure 4.

**Impact of $\rho$.** A larger $\rho$ indicates retaining more instances in the reference data pool. If the $\rho$ value is too large, it does not serve as a selection procedure and might include more noisy instances

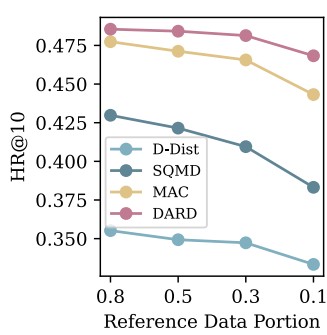

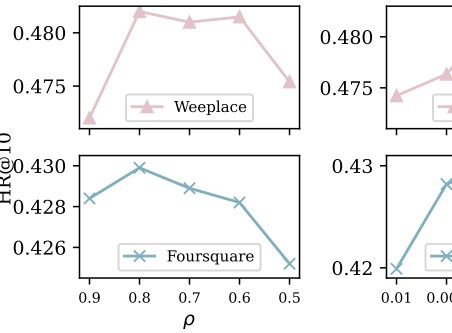

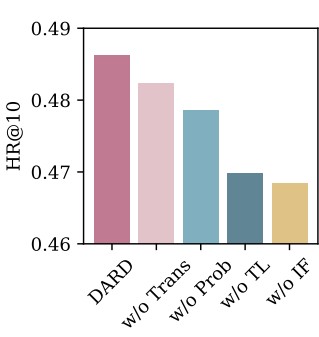

**Figure 3: The performance with different amounts of reference data on Weeplace.**

**Figure 4: The performance of DARD with loss tracking parameter $\rho$ and influence function selection parameter $\alpha$.**

**Figure 5: Result of ablation experiment on different parts of DARD on Weeplace.**

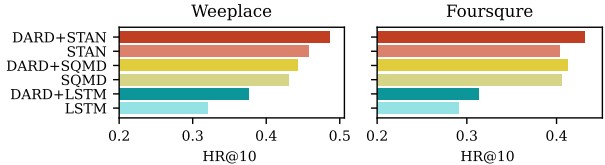

**Figure 6: Performance of DARD integrated with different Recommendation Models.**

hampering the influence function's performance post-convergence. A $\rho$ range of 0.8 to 0.6 delivers relatively satisfactory results. However, when $\rho$ is overly small, fewer instances are utilized for both model training and influence function selection, which may reduce the performance.

**Impact of $\alpha$.** A smaller $\alpha$ means that more instances are identified as harmful by the influence function. Optimal results are achieved when $\alpha$ lies between 0.01 and 0.001, as more harmful instances are excluded. However, as $\alpha$ approaches 0.0001, the performance starts to decline. A potential reason might be the misclassification of beneficial instances as harmful due to estimation errors from the influence function.

## 5.5 Ablation Study (RQ4)

In this section, we aim to demonstrate the effect of transformation generation, probability generation, data selection via training loss, and data selection via influence function. As shown in Figure 5, we implement DARD without one component, while keeping other components unchanged. we conduct experiments on Weeplace while similar trends are observed with datasets.

*w/o Trans* discards the transformation generation method and generates check-in sequences by randomly changing the items on the private reference data to preserve privacy. The performance decreases because randomly changing items cannot generate a meaningful and accurate reference data candidate pool.

*w/o Prob* discards the probability method and generates category sequences randomly. Similarly, random generation cannot guarantee a proper candidate pool.

*w/o TL* deletes the process of identifying noisy reference data instances during the training, which unavoidably introduces excessive noisy data, and impedes the effectiveness of the influence function after convergence.

*w/o IF* ignores the process of selection instances by influence function. One possible explanation for the performance decrease is that using training loss alone might not be sufficient for data selection.

## 5.6 Model-agnostic Study (RQ5)

The objective of this section is to determine whether the proposed DARD framework can seamlessly integrate with other conventional recommendation models (e.g., LSTM) and decentralized CL models (e.g., SQMD). The comparative outcomes are displayed in Figure 6. A careful analysis of these results reveals the following insights:

- The model-agnostic nature of DARD permits its compatibility with classical recommendation models, serving as the on-device model for users. This compatibility is attributed to the reference data mechanism that facilitates knowledge exchange even between heterogeneous models. Both DARD+LSTM and DARD+STAN exhibit superior performance compared to their standalone counterparts, validating the efficacy of the proposed DARD framework.
- Furthermore, the model-agnostic DARD can be seamlessly incorporated into other decentralized CL approaches that leverage reference data. The performance boost in DARD+SQMD, when contrasted with traditional CL methodologies, underscores the importance of adaptive reference data selection in enhancing knowledge exchange.

## 6 CONCLUSION

In this paper, we proposed the Decentralized Collaborative Learning with Adaptive Reference Data (DARD) framework, which allows for adaptive reference data, enhancing user collaboration and ensuring effective knowledge exchange. DARD first establishes a comprehensive yet desensitized public reference data pool, followed by collaborative pretraining and an adaptive selection of user-specific reference data, grounded in monitoring training loss and leveraging influence functions. Loss tracking identifies the noisy instances during the training, and the influence function identifies the harmful instances by estimating the difference in the model performance with and without the instances. Extensive experiments spotlight DARD's commendable performance in recommendations and its adeptness at addressing the limited reference data available.

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

## A    PRELIMINARY EXPERIMENTAL SETTINGS

To investigate whether the selection of reference data will influence the recommendation performance of a decentralized CL model. We utilize the general CL paradigm introduced in Section 4.2 on the Foursquare dataset [6]. The key characteristics of the dataset are presented in Table 1.

### A.1    Base Model and Hyper-parameters

We exploit the frequently used POI recommendation model STAN [27] as the on-device model for all users. For the base model, we follow the authors' suggestion to set the latent dimension to 50. For the CL paradigm, we follow [26] to set the neighbor number to 50, learning rate to 0.002, dropout to 0.2, batch size to 16, and training epoch to 50.

### A.2    Different Selection Strategies

To simulate the real-world situation where users are unlikely willing to share their sensitive private check-in data to the cloud or others. We utilize two data generation methods introduced in Section 4.1 on 1% users of the total dataset, locally on their own device, to generate the reference data candidate pool. Different reference data selection strategies are implemented, and the performance is evaluated by Hit Ratio@5:

- **Original**: The original candidate pool is taken as the reference data for all users.
- **Random**: The same amount of data instances are randomly selected from the candidate pool for every user collaboration.
- **Popular**: The same amount of data instances are selected from the candidate pool for every user collaboration based on the popularity of the POIs. More popular items have a higher chance of being chosen as the reference data.
- **Adaptive**: The proposed method DARD is utilized to select adaptive reference data for each user.

## B  PROOF OF LEMMA

LEMMA 2. *Discarding or downweighting the training samples in* $\mathcal{D}_- = \{\mathcal{X}_j \in \mathcal{D} | \Psi_\theta(\mathcal{X}_j) > 0\}$ *from* $\mathcal{D}$ *could lead to a model with lower test risk over* $\mathbf{Q}$:

$$L(\mathbf{Q}, \hat{\theta}_\epsilon) - L(\mathbf{Q}, \hat{\theta}) \approx -\frac{1}{m} \sum_{\mathcal{X} \in \mathcal{D}_-} \Psi_\theta(\mathcal{X}_j) \qquad (12)$$

*where* $\hat{\theta}_\epsilon$ *denotes the optimal model parameters obtained by updating the model's parameters with discarding or downweighting samples in* $\mathcal{D}_-$.

PROOF. Recall that $\hat{\theta} = \arg\min_\theta \frac{1}{m} \sum_{j=1}^m l_j(\theta)$. In this way, downweighting the training sample $\mathcal{X}_j$ in $\mathcal{D}_-$ means setting $\epsilon_j = [-\frac{1}{m}, 0)$ (Noticed that $\epsilon_i = -\frac{1}{m}$ means discarding training sample $\mathcal{X}_j$). For convenience of analysis, we set all $\epsilon_j$ equal to $-\frac{1}{m}$ and have $\Psi_\theta(\mathcal{X}_j) \triangleq \sum_{k=1}^{m'} \Psi_\theta(\mathcal{X}_j, \mathcal{X}_k^c)$. According to Eq. (10), we can estimate how the test risk is changed by discarding or downweighting $\mathcal{X}_j \in \mathcal{D}_-$ as follows:

$$L(\mathbf{Q}, \hat{\theta}_\epsilon) = \sum_{\mathcal{X}_j \in \mathcal{D}_-} \sum_{j=1}^{m'} l(\mathcal{X}_k^c, \hat{\theta}_{\epsilon_j}) - l(\mathcal{X}_k^c, \hat{\theta})$$

$$\approx \sum_{\mathcal{X}_j \in \mathcal{D}_-} \epsilon_j \times \sum_{j=1}^{m'} \Psi_\theta(\mathcal{X}_i, \mathcal{X}_k^c) \qquad (13)$$

$$= -\frac{1}{m} \sum_{\mathcal{X}_j \in \mathcal{D}_-} \Psi_\theta(\mathcal{X}_j) \leq 0$$

□

## C  GENERAL CL-BASED RECOMMENDATION DETAILS

In this section, we introduce the details of general CL-based recommendations.

### C.1  Neighbor Identification

In the DARD approach, each user's local models are trained using their own check-in sequences. However, limited check-ins on individual user devices can hinder the development of a precise POI recommender. To address this, DARD enables each user device $u_i$ to share knowledge with its neighbors. These neighbors are users who have high similarity to $u_i$, namely geographical neighbors $\mathcal{G}(u_i)$ and semantic neighbors $\mathcal{S}(u_i)$. The set $\mathcal{G}(u_i)$ includes users within the same region $r$, which is the most recent region that user $u_i$ visited.

Semantic neighbors are users that are seen as highly related if the POIs they visit belong to the same categories [19]. This relevance holds even if the users are far apart geographically. We measure this semantic similarity using category-based user preferences. Specifically, we represent each user's category distribution with $CP(u_i) = \{\mathcal{P}(c_1), \mathcal{P}(c_2), ...\mathcal{P}(c_{|C|})\}$, derived from $\mathcal{X}^c(u_i)$. We use Kullback-Leibler (KL) divergence [7] to calculate the difference between two users' category preferences:

$$d_{cat}(u_i, u_j) = KL\left(CP(u_i) \,||\, CP(u_j)\right). \qquad (14)$$

For user $u_i$, the $h$ users with the smallest values of $d_{cat}(u_i, u_j)$ are chosen as semantic neighbors.

