# OpenReview forum: "Decentralized Collaborative Learning with Adaptive Reference Data for On-Device POI Recommendation"
_ACM.org/TheWebConf/2024/Conference — TheWebConf24 Oral_

### Official Review · Reviewer_qXQA · 2023-11-21

**Novelty:** 7
**Technical Quality:** 6

**Review:**

This paper proposes an innovative approach, Decentralized Collaborative Learning with Adaptive Reference Data (DARD), to enhance the collaborative learning paradigm in on-device Point-of-Interest (POI) recommendation systems. DARD challenges the conventional method of using uniform reference data for collaboration by introducing a decentralized framework that adapts reference data to the specific context of each user pair. This approach aims to increase the efficiency and accuracy of collaborative learning in location-based social networks.
Pros:
1.	DARD introduces a novel framework for adaptive reference data in collaborative learning, addressing the limitations of existing methods that employ a one-size-fits-all approach for reference data in user collaborations.
2.	The paper presents a robust experimental setup with five distinct experiments. These cover aspects such as recommendation performance, resilience against limited reference data, and a hyperparameter study, providing a thorough validation of the proposed method.
3.	The paper is well-structured and articulated, making it accessible and easy to understand. This clarity enhances the paper's contribution to the field.
4.	The proposed paradigm of selecting adaptive reference data is not only beneficial for POI recommendations but also holds potential for broader applications in fields that utilize collaborative learning with reference data.
Cons:
1.	Ambiguity in Experimental Procedure: In Section 5.3, the process of selecting candidates when the reference pool is reduced from 100% to 10% is not clearly explained. Clarification on whether this selection is random or governed by specific rules would enhance the paper's comprehensiveness.
2.	Need for Clarification in Methodology: Section 4.1 on transformation generation requires additional detail. It is unclear whether the transformation involves exchanging the last POI of two sequences or all POIs post a shared POI.

**Questions:**

1.	Can you clarify the specifics of the transformation process in Section 4.1? Is it restricted to the last POI or does it include all subsequent POIs after a shared reference point?
2.	Can you explain the criteria or methodology was used to select candidates when reducing the reference data pool in the experiments detailed in Section 5.3?

**Ethics Review Description:**

NULL

**Reviewer Confidence:**

4: The reviewer is certain that the evaluation is correct and very familiar with the relevant literature

**Scope:**

4: The work is relevant to the Web and to the track, and is of broad interest to the community

---

### Official Review · Reviewer_3nFG · 2023-11-21

**Novelty:** 6
**Technical Quality:** 7

**Review:**

A typical decentralized recommendation paradigm is collaborative learning, where users’ personal models are updated by mutually sharing their soft decisions (logits) on a public reference dataset. This paper queries the efficacy of using the same reference dataset across all users, where a preliminary experiment proves the necessity of an adaptive approach to reference data selection. This paper further proposes an adaptive reference data selection method based on loss tracking and influence functions to remove harmful reference data points from the candidate pool. In summary, this paper introduces an innovative idea for a fairly new recommendation task, and the work is overall well-motivated and clearly presented.

My summary of this paper’s strengths:
+ This paper focuses on a niche yet very interesting problem in a special decentralized POI recommendation paradigm, where the main claims are well supported by experimental analysis.
+ The proposed approach shows a good level of novelty, and is relatively easy to follow.
+ The experiments are comprehensive, showing a wider coverage of analyses conducted. The results have also been able to support the efficacy of using adaptive reference data points for decentralized POI recommendation.

My summary of this paper’s weaknesses:
- The scope of the recommendation setting can use some further elaborations, especially for its generalizability. It would be useful to understand whether or not this approach is only applicable to POI recommendation or also applicable to general sequential recommendation, given the need for identifying neighbor clients.
- While this paper mostly discusses how to select quality reference data, it is also interesting to see the connection between how the quality of the initial candidate pool affects the final reference dataset, including its size and diversity.
- Some expressions can be made clearer. For example, “with only minor performance drops from 0.8 to 0.3” in line 785 seems to indicate that the performance drop is minor from the ratio of 0.8 to the ration of 0.3 (instead of the actual HR@10 performance), which is misleading at the first place.

**Questions:**

(1) Is the proposed approach also compatible with general sequential recommender systems?
(2) It was stated that “5% users are randomly selected to generate desensitized check-in data with the same amount of local private data”. Is this for generating candidate reference data points? Some details are needed regarding the scale of the generated data points. Do they vary a lot during five different runs?
(3) Does the Step 4 retraining require the same number of epochs as the Step 2 training (both use N in Alg 1)?

**Reviewer Confidence:**

4: The reviewer is certain that the evaluation is correct and very familiar with the relevant literature

**Scope:**

4: The work is relevant to the Web and to the track, and is of broad interest to the community

---

### Official Review · Reviewer_TnVB · 2023-11-23

**Novelty:** 6
**Technical Quality:** 7

**Review:**

This paper addresses the privacy concerns and inefficiency of centralized cloud-based systems by proposing an on-device POI recommendation approach. DARD focuses on customizing reference data for individual users, thereby enhancing personalization and preserving privacy. The framework uses innovative techniques like knowledge distillation and adaptive reference data selection to overcome the limitations of uniform reference datasets and to accommodate diverse user interests and device capabilities. The paper emphasizes the importance of this approach in the context of evolving privacy regulations and varied user data distributions. Experiments highlight DARD’s superiority in recommendation performance and addressing the scarcity of available reference data.

Pros:
1. The DARD framework with loss tracking during training and influence function post-training is a commendable innovation. These two steps are interdependent and collaboratively contribute to the identification of non-contributory instances.
2. The collaborative learning paradigm with adaptive reference data is practical, which mitigates privacy concerns that are inherent in centralized models containing comprehensive user-item interactions and user behaviours.
3. The paper effectively outlines the progression and existing limitations in the field, setting a solid foundation for the introduction of the proposed methods. This context provides clarity on the relevance and significance of the research.

Cons:
1. The description of the generation process for the Reference Data Candidate Pool in Section 4.1 needs to be detailed. A more detailed technical explanation would give a clearer understanding of the methodology.
2. There is an inconsistency in the formatting of equations throughout the paper. Some end with commas, while others, such as Equation (9), lack punctuation. Uniformity in presentation would enhance the paper's professionalism and readability.

**Questions:**

1. Could you provide more detail on the process involved in generating the reference data candidate pool, particularly regarding the transformation aspect?
2. What is the significance of retaining category information for each POI within the framework? How does this information contribute to the effectiveness of the proposed method?

**Ethics Review Description:**

N.A.

**Reviewer Confidence:**

4: The reviewer is certain that the evaluation is correct and very familiar with the relevant literature

**Scope:**

4: The work is relevant to the Web and to the track, and is of broad interest to the community

---

### Official Review · Reviewer_4sYp · 2023-11-27

**Novelty:** 6
**Technical Quality:** 7

**Review:**

This paper concentrates on enhancing on-device Point-of-Interest (POI) recommendations within Location-based Social Networks. It critically addresses the inefficiency of employing uniform reference data for collaborative interactions among users with diverse preferences and local data distributions. The authors introduce the Decentralized Collaborative Learning with Adaptive Reference Data (DARD) framework, innovatively designed to customize intermediate reference data in accordance with two users in the collaboration. This customization aims to optimize the efficacy of Collaborative Learning. Experimental results on two datasets show the effectiveness of the proposed method.

## Pros:
1.	Rational Framework: The decentralized collaborative learning framework is sound for POI recommendations, where the location and the user preferences privacy are vital, and it is rational to deploy the recommendation on the device side and enhance the model training with collaborative learning.

2.	Clear Motivation and Gap Identification: Since the way of knowledge exchange between on-device recommendations is through reference data. Using the same data for collaboration between different users is suboptimal and a new design approach is expected.

3.	Concreteness of Methodology: The proposed DARD method is thoroughly and clearly elaborated, showcasing a well-structured and concrete framework.

4.	Comprehensive Experiments: The experiments conducted are extensive and effectively demonstrate the efficacy of the proposed methods, reinforcing the paper's findings.

## Cons:

1.	Terminology Clarity: The paper's use of “harmful” and “noisy” instances interchangeably is somewhat confusing. A more distinct definition or clarification of these terms would be beneficial, especially if they both imply data instances that are redundant and detrimental to the training process.

2.	Rationale for Loss Tracking: The paper should emphasize more on the necessity of adding loss tracking during the training phase. Given that the influence function post-training can identify non-contributing data instances and construct user-specific adaptive reference data, the additional value of loss tracking during training could be further elucidated.

**Questions:**

1.	What is the difference between “harmful” instances and “noisy” instances?

2.	What is the rationale behind identifying non-contributing instances during training through loss tracking, especially when the influence function can achieve similar outcomes post-training?

**Reviewer Confidence:**

4: The reviewer is certain that the evaluation is correct and very familiar with the relevant literature

**Scope:**

4: The work is relevant to the Web and to the track, and is of broad interest to the community

---

### Decision · Program_Chairs · 2024-01-22

**Decision:**

Accept (Oral)

**Comment:**

Quality:
 + Well-written paper.
 + Sound approach/framework.

 Clarity:
 + Writing is very clear. The proposed DARD method is thoroughly and clearly elaborated.
 + The approach is well-motivated.
 + The paper is well-positioned within the existing literature.
 - Some minor and easy-to-resolve issues with punctuation and uniformity in presentation are identified.

 Originality:
 + The proposed DARD framework with loss tracking during training and influence function post-training is innovative.

 Significance:
 + The contribution is well suited for WEB and of interest to a recommender systems audience.